# German medical students´ views regarding artificial intelligence in medicine: A cross-sectional survey

Stuart McLennan[1]*, Andrea Meyer[2], Korbinian Schreyer[1], Alena Buyx[1]

1 Institute of History and Ethics in Medicine, TUM School of Medicine, Technical University of Munich, Munich, Germany, 2 Department of Psychology, Division of Clinical Psychology and Epidemiology, University of Basel, Basel, Switzerland

* stuart.mclennan@tum.de

## Abstract

### Background

Medical students will likely be most impacted by the envisaged move to artificial intelligence (AI) driven digital medicine, and there is a need to better understand their views regarding the use of AI technology in medicine. This study aimed to explore German medical students´ views about AI in medicine.

### Methods

A cross-sectional survey was conducted in October 2019 with all new medical students at the Ludwig Maximilian University of Munich and the Technical University Munich. This represented approximately 10% of all new medical students in Germany.

### Results

A total of 844 medical students participated (91.9% response rate). Two thirds (64.4%) did not feel well informed about AI in medicine. Just over a half (57.4%) of students thought that AI has useful applications in medicine, particularly in drug research and development (82.5%), less so for clinical uses. Male students were more likely to agree with advantages of AI, and female participants were more likely to be concerned about disadvantages. The vast majority of students thought that when AI is used in medicine that it is important that there are legal rules regarding liability (97%) and oversight mechanisms (93.7%), that physicians should be consulted prior to implementation (96.8%), that developers should be able to explain to them the details of the algorithm (95.6%), that algorithms should use representative data (93.9%), and that patients should always be informed when AI is used (93.5%).

### Conclusions

Medical schools and continuing medical education organisers need to promptly develop programs to ensure that clinicians are able to fully realize the potential of AI technology. It is also important that legal rules and oversight are implemented to ensure that future clinicians

**Data Availability Statement:** All data are in the manuscript and/or Supporting information files.

**Funding:** The authors received no specific funding for this work.

**Competing interests:** The authors have declared that no competing interests exist.

are not faced with a workplace where important issues around responsibility are not clearly regulated.

## Author summary

Medical students will likely be most impacted by the expected move to AI-driven digital medicine and there is a need to better understand their views regarding the topic. Germany offers a useful setting to examine these issues due to the number of AI initiatives in healthcare and the recently enacted Digital Healthcare Act. In contrast to previous studies that have typically used small online samples and focused on radiology, we used a paper-based survey and focus on Munich medical students´ general views about AI in medicine, and achieved a very high (92%) response rate. We found that medical students had relatively low awareness regarding AI in medicine and that the issue of responsibility was a major concern for them. These findings support calls for education programs to be developed to ensure that clinicians are able to fully realize the potential of AI technology. It is also important that legal rules and oversight are implemented to ensure that future clinicians are not faced with a workplace where important issues around responsibility are not clearly regulated.

## Introduction

Advances in computing processing power, the availability of digital "big data", and the development of more sophisticated algorithms, have created significant opportunities around artificial intelligence (AI). Although there is currently no established definition of what constitutes AI, the term has traditionally referred "to any machine or algorithm that is capable of observing its environment, learning, and based on the knowledge and experience gained, take intelligent actions or propose decisions" [1]. AI technologies are already widely used in everyday applications, and it is expected that they will increasingly lead to important changes in society and the economy [1].

The implementation of AI applications is particularly expected to transform healthcare [2–3]. With the ability to learn from large sets of clinical data, medical AI applications have the potential to support a range of activities, including diagnosis [4], clinical decision making [5], personalized medicine [6], therapy [7], clinical research [8], drug development [9], and administrative processes [10]. Additionally, some AI applications are responsive to the patient and their environment through a physically embodied presence, such as artificially intelligent robotic agents or smart prostheses [7,11–12]. It is envisaged that such AI applications will help improve the quality and effectiveness of care, control expenditure, reach underserved or vulnerable populations, and relieve overstretched healthcare services [13–15].

AI technology, however, also raises important ethical challenges around privacy, data protection, transparency and explainability, bias, autonomy and responsibility, and the impact of automation on employment [16–19]. In healthcare settings, these issues take on particular importance given AI applications may be interacting directly with patients in various states of vulnerability [7,20–21]. Ethical concerns around AI technology in general have prompted a rush towards "AI Ethics" to consider how AI technology can be developed and implemented in an ethical manner [22–25]. In healthcare, the WHO´s recently published guidance document "Ethics and Governance of Artificial Intelligence for Health" identifies six key ethical

principles for the use of AI in health: 1) protecting autonomy; 2) promoting human safety and well-being; 3) ensuring transparency; 4) fostering accountability; 5) ensuring equity; and 6) promoting tools that are responsive and sustainable [26].

With AI technology poised to have a significant impact on medical practice, attention is increasingly focusing on how the healthcare work force can be prepared for this change [27–28], and thus on exploring the views of current physicians [29–30] and medical students [29,31–34]. As future clinicians entering the clinic during intense technological development and change, medical students will likely be most impacted by the expected move to AI-driven digital medicine. However, previous research involving medical students is to date limited. In 2019, Oh and colleagues reported the results of an online survey conducted with a sample of 669 participants (including 121 medical students) associated with a South Korean university to assess physicians´ awareness and attitudes towards medical AI [29]. Familiarity with AI among participants overall was very low (6%), but most participants considered AI useful in the medical field (73%). Other studies involving medical students in Germany [31], Switzerland [32], Canada [33], and the United Kingdom [34], have primarily focused on the use of AI in radiology. In 2019, dos Santos and colleagues reported the finding of an online survey with 263 undergraduate medical students at three major German universities; it was found that just over half (52%) of participants were aware that AI is an important issue in radiology, and that male and tech-savvy participants were more confident regarding the benefits of AI and less concerned of these technologies [31]. In 2019, van Hoek and colleagues reported the findings of an online survey with 170 participants (including 55 medical students) throughout the German speaking part of Switzerland; it was found that the majority of participants agreed that AI should be included as a support system in radiology, but medical students saw a potential threat of AI as more likely than radiologists did, and 26% of the students that did not intend to specialize in radiology stated that AI was one of the reasons [32]. Gong and colleagues also reported in 2019 the results of an online survey conducted with 322 medical students at all 17 Canadian medical schools; it was found that only a minority (29%) of students agreed that AI would replace radiologists in foreseeable future, but a majority (68%) agreed AI would reduce the demand for radiologists [33]. In 2020, Sit and colleagues reported the results of an online survey conducted with 484 medical students from 19 UK medical schools; it was found that a majority (88%) of the students believed that AI will play an important role in healthcare, and just under half (49%) of the students reported they were less likely to consider a career in radiology due to AI [34].

There is a need to better understand medical students' views regarding the use of AI technology in medicine. Germany offers a useful setting to examine these issues. Germany, and the state of Bavaria in particular, is at the forefront of AI research and development in Europe, and has a number of AI initiatives in healthcare and medicine [35–36]. Furthermore, Germany enacted a new Digital Healthcare Act in 2019 that allows digital health applications to be prescribed and reimbursed under statutory health insurance [37–38]. With nearly 90% of the German population covered by statutory health insurance, it is expected that this will serve as an additional push towards translating AI health technology into clinical practice and the doctor-patient relationship [39].

Whereas previous studies regarding medical students´ views regarding AI in medicine have used small online samples and have typically focused on radiology, this study will take a different approach and use a paper-based survey and focus on medical students´ general views about AI in medicine. In particular, this study aims to explore medical students´: 1) knowledge of AI in medicine, 2) views about its usefulness in medicine, 3) views about what should happen when the physician's judgment and the results of an AI algorithm differ regarding a treatment decision, 4) views about the advantages and disadvantages of AI in medicine, 5)

perceived issues with the implementation of AI in medicine, and 6) whether there are associated differences between medical student´s views about the advantages and disadvantages of AI and their self-reported gender or digital skills and competences.

## Material and methods

This study was approved by the Technical University of Munich´s Research Ethics Committee on 25 September 2019 (approval no. 433/19 S). All participants signed an informed consent form.

### Survey implementation

A cross-sectional survey was conducted on 14 October 2019. All new medical students at the Ludwig Maximilian University of Munich and the Technical University Munich are required to complete a joint compulsory course ("Berufsfelderkundung") in the first two and a half days of their first semester. The course provides a career exploration for students and an overview of the fields of medicine; there is no specific focus on AI. A total of 918 students were registered for the compulsory course in 2019. This represents approximately 10% of all new medical students in Germany [40]. In the context of this course, students were invited to participate in the study; it was stressed that their participation in the study was voluntary and completely independent of the successful completion of the course. Students were provided the paper-based study materials in the lecture theatres and given time to fill in the survey. The survey was prepared using the automation software EvaSys (EvaSys Central Evaluation version 8.0). Completed surveys were later scanned and data automatically entered into the software. Missing values were checked against scans of the original completed survey. If an answer had not been picked up by EvaSys the value was entered into SPSS. If there was no answer, it was left as a missing value. Data were then exported into Statistical Package for the Social Sciences (SPSS version 24 for Windows, IBM Corporation).

### Survey contents

Survey questions were developed by the research team, informed by the review of previous surveys conducted with medical students and physicians [29–34]. The survey was conducted in German, and has been translated into English using back translation (see S1 Appendix). Questions covered key ethical principles identified for the use of AI in health, such as the six principles described in the WHO´s recently published guidance document "Ethics and Governance of Artificial Intelligence for Health" [26]. The survey was pilot-tested with a convenience sample of five students to ensure clarity and item comprehension; no changes to the survey were required. The pilot test was done separately and the results are not included in the present study. Survey questions explored participants´ 1) knowledge of AI and views about its usefulness in medicine, 2) views about what should happen when the physician's judgment and the judgment of an AI algorithm differ regarding a treatment decision, 3) views about the advantages of AI in medicine, 4) views about the disadvantages of AI in medicine, and 5) implementation issues. Participants were instructed to read the statements and indicate on a 9-point Likert scale (from "I do not agree at all" to "I completely agree") the extent they agree or disagree with the statements. However, due to a human error uploading the questions to the EvaSys platform, questions in the first section regarding respondents' knowledge of AI and views about its usefulness in medicine were put on a 7-point scale. A "no answer" option was not provided. Demographic questions asked for participants' age and gender. All data were collected in a pseudonymous form.

## Data analysis

Descriptive statistics included absolute and relative frequencies per category and, in order to describe central tendencies, medians and interquartile ranges. When reporting absolute and relative frequencies, the majority proportion of participants who agreed or disagreed with the statement is reported; the remaining participants were either neutral (4 in the case of 7-point and 5 in the case of 9-point scale) or agreed/disagreed. All answers were included; no participant was excluded for not answering enough questions. There were no inconsistent answer patterns. Tests for differences between females and males regarding agreement about the advantages and disadvantages of AI are based on Welch's t-tests for independent samples (due to unequal sample sizes between females and males), and tests for associations between digital skills and competences and agreement about the advantages and disadvantages of AI are based on Spearman's rank correlation. Spearman's rank correlation coefficient was used since it does not assume normally distributed variables which would be unrealistic for items measured on a Likert scale. Cohen's d was calculated to measure the magnitude of the effect size. Control for multiple testing was deliberately omitted due to the explorative nature of the study. All statistical analyses were conducted using SPSS version 24.0 (IBM Corp. Armonk, NY, USA).

## Results

### Characteristics of participants

A total of 844 students participated, corresponding to a 91.9% (844/918) response rate. Overall, 66.8% (545/844) of participants identified as female, and 33.1% (270/844) identified as male. One participant (1/844; 0.12%) identified as other, and the gender of 28 participants was missing. Participants had a median age of 19, ranging from 18 to 52 years old.

### Artificial intelligence knowledge and perceived usefulness in medicine

Almost three quarters (610/838; 72.8%) of participants agreed that they had good digital skills and competences. However, about two thirds of them (539/838; 64.3%) did not feel they were well informed about AI in medicine (Table 1 and S1 Table).

The majority of participants (463/807; 57.4%) agreed that AI has especially useful applications in the medical field, agreeing that AI will be particularly useful in improvement of drug research and development (688/834; 82.5%), supporting physicians making a diagnosis (522/

**Table 1. Artificial intelligence knowledge and perceived usefulness in medicine.**

| Question | N | Median, Mean[a] | Inter-quartile range |
|---|---|---|---|
| Overall, I have good digital skills and competences | 838 | 5, 5.1 | 2 |
| I feel well informed about AI in medicine | 838 | 3, 3.1 | 2 |
| AI has especially useful applications in the medical field | 807 | 5, 4.7 | 2 |
| In what area of medicine will AI be particularly useful? | | | |
| Supporting physicians making a diagnosis | 834 | 5, 4.7 | 3 |
| Supporting physicians in making treatment decisions | 837 | 5, 4.4 | 3 |
| Supporting physicians directly during treatment | 835 | 5, 4.5 | 3 |
| Patients treating themselves independently with AI health apps | 836 | 2, 2.4 | 2 |
| Improvement of drug research and development | 834 | 6, 5.5 | 1 |
| Supporting personalised medicine | 836 | 5, 4.7 | 2 |

[a] Due to an error uploading the questions to the EvaSys platform, questions in the first section of the survey were put on a 7-point scale: 1 = I do not agree at all—7 = I completely agree.

**Table 2. Conflicts between judgement of physician and AI.**

| Question | N | Median, Mean[a] | Inter-quartile range |
|---|---|---|---|
| If the physician's judgment and the judgment of a well-tested and accurate AI algorithm differ with regard to a treatment decision, then. . . | | | |
| . . .the judgement of the physician should be followed | 833 | 7, 7.1 | 2 |
| . . .the judgement of the algorithm should be followed | 833 | 3, 3.1 | 2 |
| . . .the patient should choose which judgement should be followed | 823 | 4, 4.0 | 5 |

[a] Scale: 1 = I do not agree at all—9 = I completely agree

834; 62.6%), supporting personalised medicine (494/836; 59.1%), supporting physicians directly during treatment (491/835; 58.8%), and supporting physicians in making treatment decisions (455/837; 54.4%). However, the vast majority of participants disagreed (678/836; 81.1%) that patients will treat themselves independently with AI health apps.

## Conflicts between judgement of physician and AI

A large majority (687/833; 82.5%) of participants agreed that the judgement of the physician should be followed if the judgement of the physician conflicts with the results of a well-tested and precise AI algorithm (Table 2 and S2 Table). Only a small minority of participants agreed that the results of the AI algorithm should be followed (70/833; 8.4%), or that the patient should choose which judgement should be followed (251/823; 30.5%). Participants who agreed more strongly that the judgement of the physician should be followed in conflicts, less strongly agreed that the judgement of the AI algorithm should be followed (r = –0.79).

## Advantages and disadvantages of AI in medicine

The vast majority of participants agreed that the possible advantages of using AI in medicine include the analysis of large amounts of clinically relevant data (794/832; 95.4%), and giving physicians more time for discussions and clinical examinations (764/823; 92.8%) (Table 3 and S3 Table). A large majority also agreed that a possible advantage of AI is the reduction of medical errors (733/825; 88.8%), and that AI does not get tired and can work 24 hours (660/827; 79.8%). Around two thirds of participants also agree that it could improve the cost efficiency of medicine (566/825; 68.6%), and making more precise treatment decisions (528/831; 63.5%). However, there were some significant differences between genders and participants´ perceived digital skills and competences (S4 Table). Male participants were significantly more likely to

**Table 3. Advantages of AI in medicine.**

| Question | N | Median, Mean[a] | Inter-quartile range |
|---|---|---|---|
| I find the following possible advantages of using AI in medicine important | | | |
| Analysis of large amounts of clinically relevant data | 832 | 8, 8.0 | 2 |
| Making more accurate treatment decisions | 831 | 6, 5.8 | 3 |
| Reducing medical errors | 825 | 7, 7.2 | 1 |
| Improving the cost efficiency of medicine | 825 | 7, 6.3 | 3 |
| Giving physicians more time for discussions and clinical examinations | 823 | 8, 7.7 | 2 |
| AI does not get tired and can work 24 hours | 827 | 8, 7.0 | 3 |

[a] Scale: 1 = I do not agree at all—9 = I completely agree

agree that possible advantages of medical AI include analysis of large amounts of clinically relevant data (t(558) = −2.5, $P$ = 0.01, d = 0.21), making more precise treatment decisions (t(532) = −3.9, $P$ < .001, d = 0.34), the reduction of medical errors (t(623) = −4.1, $P$ < .001, d = 0.33), and that AI does not get tired and can work 24 hours (t(560) = −3.3, P < .001, d = 0.28). Effect sizes for these gender differences varied between small and small to medium (0.21–0.34). Similarly, participants who agreed more strongly that they had good digital skills and competences also agree more strongly that possible advantages include the analysis of large amounts of clinically relevant data (r = 0.19, $P$ < .001), making more precise treatment decisions (r = 0.08, $P$ = 0.02), and the reduction of medical errors (r = 0.12, $P$ < .001); however, effect sizes were generally small (.08, .12, and .19 respectively).

The vast majority of participants agreed that the possible disadvantages of using AI in medicine include the lack of ability to develop empathy and to take into account the emotional well-being of the patient (735/808; 90.9%) (Table 4 and S5 Table). A large majority also agreed that a possible disadvantages of AI is that it cannot be used for advice in unforeseen situations due to insufficient information (700/815; 85.9%), that it can cause uncertainty as to who is liable in the event of an error (666/813; 81.9%), that it can undermine the autonomy of physicians (647/808; 80.1%), that it is not flexible enough to be used with every patient (641/816; 78.5%), and that it can be developed by programmers with little experience in medical practice (603/814; 74.1%). Around two thirds of participants also agreed that AI can amplify biases that already exist in data sets and use them to discriminate against patients (526/812; 64.8%), and that it can undermine the autonomy of patients (494/814; 60.7%). However, there were some significant differences between genders and participants´ perceived digital skills and competences (S6 Table). Female participants were significantly more likely to agree that possible disadvantages of medical AI include that it cannot be used for advice in unforeseen situations due to insufficient information (t(430) = 2.6, $P$ = .008, d = 0.25), that it is not flexible enough to be used with every patient (t(409) = 7.1, $P$ < .001, d = 0.70), that it can undermine the autonomy of patients (t(465) = 2.7, $P$ = .006, d = 0.25), that it can undermine the autonomy of physicians (t(437) = 3.3, $P$ = .001, d = 0.32), the lack of ability to develop empathy and to take into account the emotional well-being of the patient (t(407) = 3.7, $P$ < .001, d = 0.37), and that it can cause uncertainty as to who is liable in the event of an error (t(436) = 2.5, $P$ = .01, d = 0.24). Effect

**Table 4. Disadvantages of AI in medicine.**

| Question | N | Median, Mean[a] | Inter-quartile range |
|---|---|---|---|
| How important do you consider the following possible disadvantages of using artificial intelligence in medicine? | | | |
| Cannot be used for advice in unforeseen situations due to insufficient information | 815 | 8, 7.3 | 2 |
| Not flexible enough to be used for every patient | 816 | 7, 6.9 | 3 |
| Can amplify biases that already exist in data sets and lead to patient discrimination | 812 | 7, 6.0 | 3 |
| Can undermine the autonomy of patients | 814 | 6, 5.9 | 4 |
| Can undermine the autonomy of physicians | 808 | 7, 6.8 | 2 |
| The lack of ability to develop empathy and consider the patient´s emotional well-being | 808 | 9, 7.9 | 1 |
| Can be developed by programmers with little experience in medical practice | 814 | 7, 6.7 | 3 |
| Causes uncertainty as to who is liable in the event of an error | 813 | 8, 7.1 | 3 |

[a] Scale: 1 = I do not agree at all—9 = I completely agree

sizes for these gender differences varied between small and small to medium (0.24–0.37), except for flexibility of use which was medium to large (0.70). Furthermore, participants who agreed more strongly that they had good digital skills and competences less strongly agreed that possible disadvantages include that AI can undermine the autonomy of patients (r = −0.12, $P$ < .001), however, the effect size was small.

## Implementation issues

The vast majority of the participants agreed that when AI is used in medicine it is important that there are legal rules to clarify liability in case of an error (753/776; 97%), that physicians were consulted prior to the implementation of an AI algorithm system in clinical practice (776/802; 96.8%), that the developers can explain the rules and parameters of the algorithm to physicians (768/803; 95.6%), that the underlying data of the algorithms are representative (752/801; 93.9%), that oversight mechanisms are in place to evaluate the performance of an AI algorithm in clinical practice (726/775; 93.7%), and that patients are always informed when AI algorithms are used (737/788; 93.5%) (Table 5 and S7 Table). A large majority also agreed that physicians should have a choice whether to use an AI algorithm (646/790; 81.8%), and that patients should have a choice whether AI algorithms are used in their treatment (640/782; 81.8%). About two thirds agreed that the topic of AI should receive a lot of attention in medical studies (514/776; 66.2%), while slightly more than half agreed that patients should have equal access to the system (394/779; 50.6%).

## Discussion

It appears that this is one of the first studies to explore medical students´ general views about AI in medicine. It has also used one of the biggest and clearly defined samples of medical students to date, and achieved a very high (92%) response rate. Key results of this study include: (1) two thirds of medical students did not feel well informed about AI in medicine, (2) just over a half of students thought that AI has useful applications in medicine, but were more likely to identify the potential of AI to help improve drug research and development compared

**Table 5. Implementation issues.**

| Question | N | Median, Mean[a] | Inter-quartile range |
|---|---|---|---|
| When AI is used in medicine, it is important... | | | |
| That the underlying data of the algorithms are representative | 801 | 8, 7.9 | 2 |
| That the developers can explain the rules and parameters of the algorithm to physicians | 803 | 9, 8.2 | 1 |
| That physicians were consulted before introducing the AI algorithm system into clinical practice | 802 | 9, 8.3 | 1 |
| That physicians have a choice whether to use an AI algorithm | 790 | 8, 7.2 | 3 |
| That patients are always informed when AI algorithms are used | 788 | 9, 8.2 | 1 |
| That patients have a choice whether AI algorithms are used in their treatment | 782 | 8, 7.3 | 3 |
| That patients have equal access to the system | 779 | 6, 5.6 | 5 |
| That oversight mechanisms are in place to evaluate the performance of an AI algorithm in clinical practice | 775 | 9, 8.0 | 1 |
| That there are legal rules to clarify liability in case of an error | 776 | 9, 8.4 | 1 |
| The topic of AI should receive a lot of attention in medical studies. | 776 | 6, 6.1 | 3 |

[a] Scale: 1 = I do not agree at all—9 = I completely agree

to potential clinical uses, (3) a large majority of students thought that the judgement of the physician should be followed if the judgement of the physician conflicts with the results of a well-tested and precise AI algorithm, (4) nearly all students thought that the inability to develop empathy and to take into account the emotional well-being of the patient was a possible disadvantage, with male participants more likely to agree with advantages of AI, and female participants more likely to be concerned about disadvantages, and (5) the vast majority of students thought that when AI is used in medicine that it is important that there are legal rules regarding liability and oversight mechanism, that physicians are consulted prior to implementation and can be told the details of the algorithm, that algorithms use representative data, and that patients are always informed when AI is used.

Similar to the previous studies [29,31], this study found relatively low awareness among medical students regarding AI in medicine. Although the answers of the students participating in this study do not represent a "medical student" state, as they just started studying medicine, it is important that medical schools prepared students "for a future in which artificial intelligence is poised to play a significant role" [27]. However, there are currently limited training programs or modules in place [7,27]. A dual focused approach has been advocated, combining robust data science-focused additions to the baseline curricula with extracurricular programs to cultivate leadership in this space [27]. It will be important that medical schools and continuing medical education organisers promptly develop and implement such an approach to ensure that the healthcare work force is able to fully realize the potential of this technology. There has already been a push to improve the general ethical awareness of those in the development of AI technology, with several leading universities and research institutions now include ethics in their technical curricula with the explicit purpose of raising ethical awareness and capacities for critical reasoning in developers, programmers and engineers [41–42]. Given the ethical challenges that AI technology raises in healthcare settings, it will be crucial that curriculum for the healthcare work force implementing AI technology also includes ethical considerations.

There are great expectations for AI-driven digital medicine to elevate patient autonomy and lead to higher involvement of patients in their own care through digital self-care [43–44]. However, medical students in this study had little expectation that patients will engage in digital self-care with AI-driven medical applications. This is somewhat surprising in view of the fact that Germany is one of the world's first countries to approve digital health apps, including AI-driven apps, for prescription as part of a new law on digital medical care [38]. AI-driven apps can thus be expected to penetrate wider practice on Germany, with increasing patient use. Although this finding is in line with empirical findings on patient-driven digital self-care lagging behind expectations in expert literature [44], medical students nonetheless need to be prepared for encountering increasing numbers of highly digitally literate patients using AI-driven and other digital healthcare apps as part of the regular care and well-being regimes.

The issue of responsibility in relation to AI technology has also received widespread attention in recent years [45]. Although the participating medical students were at the very beginning of their studies, it was clear that the issue of responsibility is a major concern for them. Students were concerned about uncertainty as to who is liable in cases or error, and had strong views regarding the need for legal clarity regarding liability and for oversight mechanisms. Students also strongly agreed that the judgement of the physician should be followed in cases of conflict with AI algorithms, that patients are always informed when AI algorithms are used, and that physicians and patients should have a choice whether AI in used. These views fit nicely with current discourse on the dangers of AI confusing responsibility and undermining patients´ and physicians' ability to make autonomous informed decisions about treatment [16–21,45]. Authorities are still grappling with how to regulate and govern AI technology, and

there remains limited legal rules and oversight. It is important that this situation is addressed promptly; future clinicians should not be faced with a workplace where these important issues around responsibility are not clearly regulated. This will include not only how responsibility is allocated when things go wrong, but also how to best respect clinician judgement in learning health care systems using AI technology [46]. The general discussion around AI in medicine has shifted away from complete replacement of physicians and their judgement, to more synergistic uses of AI [31]. Nevertheless, although clinician judgement is indispensable to the practice of medicine, evidence-based medicine movement has shown that intuition and unsystematic clinical experience can be biased, up to 10 years behind latest research, and recommend ineffective therapies [46]. Not all constraints on the behaviour of clinicians will necessarily interfere with the obligation to respect clinicians' judgement, but contextual assessments of the likely impact of any proposed constraints are required [47].

## Limitations

This study has some limitations that should be taken into account when interpreting the results. The survey was conducted in only one region of Germany. Although the in-take of new medical students at the Ludwig Maximilian University of Munich and the Technical University Munich represents approximately 10% of all new medical students in Germany, there may be region-specific and country-specific differences in medical students' views about AI in medicine that might limit the ability to generalise the results to medical students in other German regions and in other countries. However, this study found similar differences between genders and participants´ digital skills and competences in relation to the perceived benefits and concerns of AI in medicine as a study conducted in western Germany [31], and given the high response rate the results should be well representative of new medical students in Munich. Nevertheless, studies focusing on medical students´ general AI views would be desirable in other German regions and countries, as would future research examining whether results may change longitudinally over time and qualitative studies to further explore and elaborate on this study's findings. As noted in the methods section, due to a human error uploading the questions to the EvaSys platform, questions in the first section of the survey were put on a 7-point scale instead of a 9-point Likert scale. As all participants responded to the items in the first section of the survey and not part of them, it is difficult to know what the results would have been had the 9-point scale been used. The study has the usual limitations of a self-reported survey: e.g. it is not known how good medical students´ digital skills and competencies are. Social desirability may have resulted in an over-reporting of knowledge regarding AI in medicine. However, this only reinforces the finding that there is low awareness among new medical students regarding this topic. There is currently no established definition of what constitutes AI, and a definition of AI in medicine was not provided in the survey, which may have been challenging for some participants. However, concrete examples of uses of AI in medicine were provided in the survey and piloting testing did not reveal any problems of understanding. Given around two thirds of participants did not feel well informed about AI in medicine, however, this may have affected their ability to answer the following questions. The survey asked participants views about general AI technology in medicine, and did not distinguish between embodied and non-embodied AI technology. It would be helpful if future studies examine whether student´s views about AI technology in medicine depend on whether the technology is embodied or non-embodied. Finally, limited demographic information (gender and age) was collected about participants because the group was very homogenous. As students progress in their studies, future studies should explore whether other demographic factors influence medical students´ views about AI.

## Supporting information

**S1 Table. Artificial intelligence knowledge and perceived usefulness in medicine.**
(DOCX)

**S2 Table. Conflicts between judgement of physician and AI.**
(DOCX)

**S3 Table. Advantages of AI in medicine.**
(DOCX)

**S4 Table. Differences among groups regarding advantages of AI in medicine.**
(DOCX)

**S5 Table. Disadvantages of AI in medicine.**
(DOCX)

**S6 Table. Differences among groups regarding disadvantages of AI in medicine.**
(DOCX)

**S7 Table. Implementation issues.**
(DOCX)

**S1 Appendix. Survey.**
(DOCX)

## Author Contributions

**Conceptualization:** Stuart McLennan, Alena Buyx.

**Data curation:** Stuart McLennan.

**Formal analysis:** Stuart McLennan, Andrea Meyer.

**Investigation:** Stuart McLennan, Korbinian Schreyer, Alena Buyx.

**Methodology:** Stuart McLennan, Alena Buyx.

**Project administration:** Stuart McLennan.

**Supervision:** Stuart McLennan, Alena Buyx.

**Writing – original draft:** Stuart McLennan.

**Writing – review & editing:** Stuart McLennan, Andrea Meyer, Korbinian Schreyer, Alena Buyx.

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
