## [Decision Letter · Decision Letter 0]

7 Jul 2022

PDIG-D-21-00121

German medical students´ knowledge and views regarding artificial intelligence in medicine: a cross-sectional survey

PLOS Digital Health

Dear Dr. McLennan,

Thank you for submitting your manuscript to PLOS Digital Health. After careful consideration, we feel that it has merit but does not fully meet PLOS Digital Health's publication criteria as it currently stands. Therefore, we invite you to submit a revised version of the manuscript that addresses the points raised during the review process.

Please submit your revised manuscript within 60 days Sep 05 2022 11:59PM. If you will need more time than this to complete your revisions, please reply to this message or contact the journal office at digitalhealth@plos.org. Please include the following items when submitting your revised manuscript:

We look forward to receiving your revised manuscript.

Kind regards,

Laura Sbaffi, PhD, MA, MSc

Section Editor

PLOS Digital Health

Journal Requirements:

1. Please amend your online Financial Disclosure statement. If you did not receive any funding for this study, please simply state: “The authors received no specific funding for this work.”

2. Please update your online Competing Interests statement. If you have no competing interests to declare, please state: “The authors have declared that no competing interests exist.”

3. Please provide a complete Data Availability Statement in the submission form, ensuring you include all necessary access information or a reason for why you are unable to make your data freely accessible. If your research concerns only data provided within your submission, please write “All data are in the manuscript and/or supporting information files.” as your Data Availability Statement.

Additional Editor Comments (if provided):

Reviewers' comments:

Reviewer's Responses to Questions

**Comments to the Author**

1. Does this manuscript meet PLOS Digital Health’s publication criteria? Is the manuscript technically sound, and do the data support the conclusions? The manuscript must describe methodologically and ethically rigorous research with conclusions that are appropriately drawn based on the data presented.

Reviewer #1: Yes

Reviewer #2: Partly

2. Has the statistical analysis been performed appropriately and rigorously?

Reviewer #1: Yes

Reviewer #2: Yes

3. Have the authors made all data underlying the findings in their manuscript fully available (please refer to the Data Availability Statement at the start of the manuscript PDF file)?

Reviewer #1: Yes

Reviewer #2: Yes

4. Is the manuscript presented in an intelligible fashion and written in standard English?

Reviewer #1: No

Reviewer #2: Yes

5. Review Comments to the Author

Reviewer #1: Please indicate the original language of the survey. And if the survey was originally in German, please briefly describe how it was translated to English.

Was the pilot test done separately? Were the results of the pilot test included in the present study? Were there any changes to the survey due to the pilot testing?

Some info on whether the Berufsfelderkundung course was related to AI or medical technology or not seems relevant to interpreting the results of this study and thus should be added to the manuscript.

The details of the scales deserve more attention than just being table footnotes. The suggestion here is that the scales could be better off described in the texts preceding the tables, and that it be made clear to the reader what the labels were and that the scales were labeled only on either end.

The term “middle category” was used in the paper without a clear definition. From the surrounding texts, it seems this meant the central part of the Likert scale. But then a clear definition is still needed, as, on a 9-point Likert scale with labels only on either end, the central part could be defined in multiple ways. Two common examples would be defining only 5 as the “middle category,” and defining 4 to 6 as the “middle category.” Also, some of the questions were done using a 7-point Likert scale, which might require a separate definition of what the “middle category” was. It can be helpful to the reader to clearly describe which points were considered “disagree,” “neutral,” and “agree.” Please clarify.

Judging from the statistical parameters used in presenting the data, the responses were likely non-normally distributed. Still, it would be helpful to clearly indicate the normality of the data to the reader.

Results from Likert scales are poorly parsable when presented in tables, especially without a frequency distribution table, and even with one it still may not help much. Please consider using another form of visual representation. As an example, the following link shows some possible ways to effectively visualize results from Likert scales. https://www.storytellingwithdata.com/blog/2012/03/visualizing-survey-data

As written by the authors, the gender differences seemed to be important findings. Clearly illustrating these differences, whether by a table or a graphic, could help in conveying the importance to the reader. Maybe there can also be further discussions regarding these differences, as there is a clear disparity between how often this topic was mentioned in the Results section and how little attention was given to it in the Discussion section. 

The results showed that most of the participants believed the physician should be trusted over well-calibrated AI in a case of conflict. In general, well-made algorithms are more precise and produce less noise than human judgment. It is thus theoretically better overall to follow the AI’s suggestions. But in the real world, there can be medico-legal reasons to lean on the physician’s side. This part seems to warrant further discussion and carries real teaching points for medical students.

The tables represent participants’ opinions on various issues, not those issues themselves. Table names should reflect this.

The fact that the participants had had no experience in medicine (they were fresh out of high school) should also be listed as a limitation of the study. In addition, “high school graduates” is probably a better description of their state than “after school.”

A space between an equal sign and a minus sign would help with legibility. This might only be a font issue and may not show up in the final manuscript, but it seems prudent to keep this in mind.

Overall, the manuscript needs to be copyedited for grammar and clarity, particularly on verb conjugation and missing articles/prepositions/punctuations. A few examples are described below.

- On page 5, the sentence starting with “In 2019,...” could use a comma between “radiologists” and “and of the…” to improve clarity. (If I’m understanding it correctly; that is quite a run-on sentence.)

- Page 10, “and” should be added before the final clause of the first and second sentences.

- Page 23, last paragraph, forewent should have been foregone.

Reviewer #2: Information content:

The manuscript describes new research that aimed to better understand medical students knowledge and views regarding the use of AI technology in medicine. It appears that this is one of the first studies to explore medical students´ general views about AI in medicine

There is a high response rate to their survey and the results of the study found relatively low awareness among medical students regarding AI in medicine and the need for curricula to ensure that clinicians are able to fully realize the potential of AI technology.

Introduction:

This section provided an overview of the relevant literature and outlined the key research questions to be addressed. 

Methods:

This section has been placed at the back of the paper after the discussion. I recommend it is brought forward following the 'Introduction' section ie in the IMRaD format.

As stated it's unfortunate that "Due to an error uploading the questions to the EvaSys platform, questions in the first section of the survey were put on a 7-point scale: 1 = I do not agree at all - 7 = I completely agree"

This should be stated in the 'limitations' section. In looking at your survey, it's unclear if student respondents would have realised the scale changed from a 7 point scale to a 9 point scale. Could you add further detail explaining whether you believe this has significantly impacted your results or not?

The study aims to assess medical students knowledge of AI - but in the questionnaire there appears to be only two questions:

 1. "Overall, I have good overall digital skills and competences"

 2. I feel well informed about artificial intelligence (AI) in medicine.

It appears the vast majority of the survey is assessing student awareness, attitudes and perceptions of AI rather than explicitly assessing knowledge of AI. This could be further acknowledged in the limitations section.

Results and discussion:

The results are clearly outlined in sufficient detail and appropriate tables have been used. The results have been discussed in relation to the relevant literature. 

Limitations of the research have been acknowledged - further limitations could be added as outlined above. Future research could also consider qualitative studies to further explore and elaborate on this study's findings. It would be also interesting to know if there are differences by year level of medical school and whether results may change longitudinally over time.

6. PLOS authors have the option to publish the peer review history of their article (what does this mean?). If published, this will include your full peer review and any attached files.

**Do you want your identity to be public for this peer review?** For information about this choice, including consent withdrawal, please see our Privacy Policy.

Reviewer #1: No

Reviewer #2: No

---

## [Editor Report · Decision Letter 1]

29 Aug 2022

German medical students´ views regarding artificial intelligence in medicine: a cross-sectional survey

PDIG-D-21-00121R1

Dear Dr. McLennan,

We are pleased to inform you that your manuscript 'German medical students´ views regarding artificial intelligence in medicine: a cross-sectional survey' has been provisionally accepted for publication in PLOS Digital Health.

Best regards,

Laura Sbaffi, PhD, MA, MSc

Section Editor

PLOS Digital Health